# A Comparison of Contributing Factors between Young and Old Riders of Motorcycle Crash Severity on Local Roads

Thanapong Champahom [1,*], Chamroeun Se [2], Sajjakaj Jomnonkwao [2], Tassana Boonyoo [3] and Vatanavongs Ratanavaraha [2]

1   Department of Management, Faculty of Business Administration, Rajamangala University of Technology Isan, Nakhon Ratchasima 30000, Thailand
2   School of Transportation Engineering, Institute of Engineering, Suranaree University of Technology, Nakhon Ratchasima 30000, Thailand
3   Traffic and Transport Development and Research Center (TDRC), King Mongkut's University of Technology Thonburi, Bangkok 10140, Thailand
*   Correspondence: thanapong.ch@rmuti.ac.th

**Abstract:** This study aims to identify the factors that influence the severity of motorcycle crashes on local roads, particularly given the high speeds often observed for motorcycles on these roads with low traffic volumes and numerous multi-leg intersections. Previous research has shown that a rider's age can impact their speed behavior. To explore this issue, data on motorcycle crashes from 2015 to 2020 in Thailand—a middle-income developing country—were analyzed using a random parameter logit model with unobserved heterogeneity in means and variances, comparing young (<30-year-old) and older (>50-year-old) riders. The contributing factors were divided into four groups: driver, crash, environmental, and road factors. The transferability test yielded different results for the young rider and old rider models, indicating that it is appropriate to analyze these models separately. A constant value revealed that old riders were more likely to die in a crash than young riders. In terms of the random parameter, the local address and road surface variables were found to be significant in both models. The results of unobserved heterogeneity in means and variances identified significant variables in both models, including gender, exceeding the speed limit, lit roads, unlit roads, mobile phone use, and road surface. These findings were used to develop policy recommendations for reducing the severity of motorcycle crashes on local roads.

**Keywords:** crash fatality; developing country; rider age; unobserved heterogeneity; random parameter logit model

## 1. Introduction

Road crashes are the leading cause of death globally, prompting significant attention and consideration as a sustainable development issue [1]. Due to detrimental effects on global health systems and economies, the United Nations has developed Targe 3.6, aiming to reduce half of road traffic deaths by 2030 as part of the Sustainable Development Goals (SDG) [2]. The highest number of crash fatalities occur in low- and middle-income countries [3], with a significant proportion being motorcycle riders [4]. As a result, many studies in these countries have focused solely on motorcycle crashes [5,6]. Thailand, a middle-income country, has high rates of motorcycle use [7,8] due to the vehicle's affordability and ability to navigate through traffic jams [9]. However, this increased use is accompanied by a high mortality rate due to the lack of protective safety measures for motorcyclists in crashes [10].

The three main road types in Thailand are distinguished by the agencies responsible for their maintenance and the risks they pose in terms of fatal accidents. The Department of Highways, Ministry of Transport maintains the highways, which connect large geographical areas, such as regions, provinces, and districts [11]. Despite the low number of motorcycle

accidents on these roads, the collisions that do occur tend to be severe due to the high speeds of drivers and the presence of large vehicles. Rural roads, which are maintained by the Department of Rural Roads, Ministry of Transport, are primarily used to connect highways to area entrances or sub-districts. These roads have a lower risk of fatal accidents due to the slower speeds and fewer trucks present. Local roads, which are maintained by local government organizations with a focus on accessibility, make up the largest percentage of roadways in Thailand at 85.58% in 2021, with a total length of 601,451 km. In comparison, highways make up 7.43% of roadways, with a total length of 52,189 km, and collector roads comprise 6.99%, with a total length of 49,124 km.

The high number of motorcycles on local roads in Thailand is a major contributor to the high risk of crashes on these roads [12]. However, the crashes that do occur on local roads are generally less severe than those on other road types. Despite the relatively small proportion of fatal crashes on local roads compared to other types of roads, it ranks second in terms of the total number of fatal crashes, following highways (Figure 1).

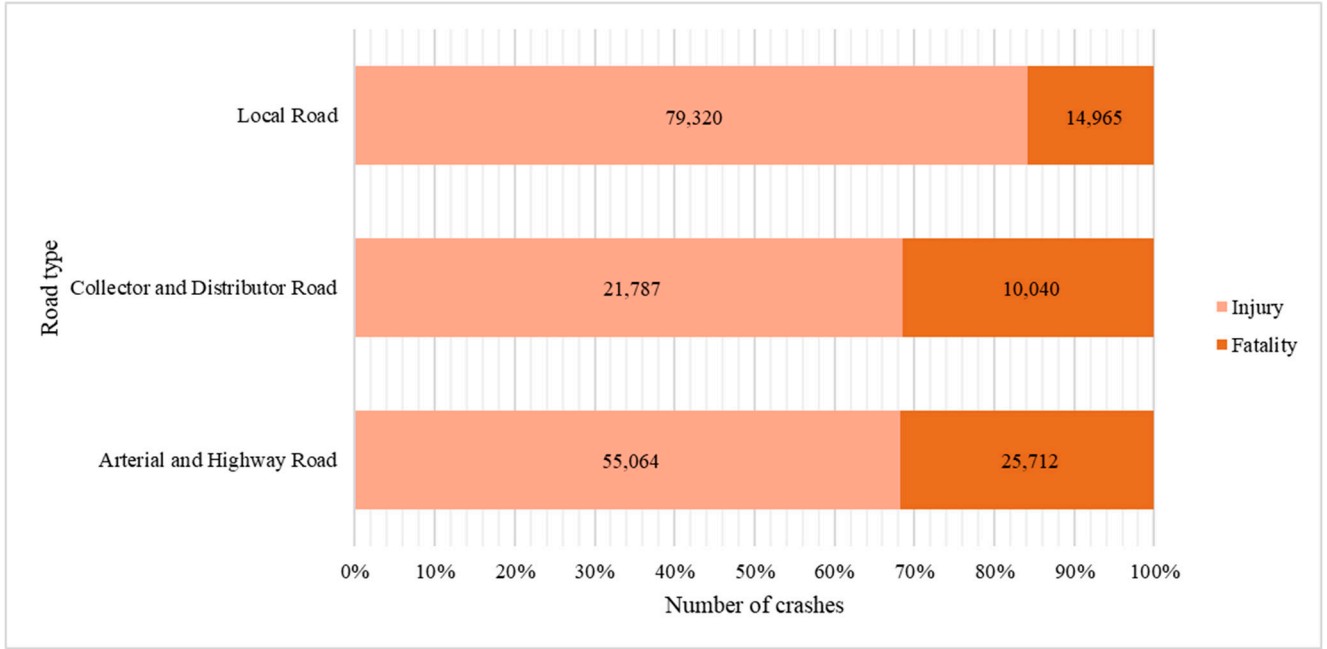

**Figure 1.** Crash severity of 2015–2020 (Comparison on road type).

Many factors can contribute to fatalities in motorcycle crashes, including the driver, the road, the vehicle, the environment, and the type of crash. Studies have shown that the age of the driver is a significant factor in deaths, especially in motorcycle crashes [13]. Older drivers, who may have less physical strength than younger drivers, are more likely to die in crashes [14]. In the context of this study, "young riders" refer to those under the age of 30, while "old riders" are defined as those over the age of 50 [15].

Table 1 presents the results of studies on the severity of motorcycle crashes, highlighting the age of the driver as a factor. Of the studies included in the table, only Islam [10] and Goodwin et al. [16] have specifically examined local roads. Both of these studies were conducted in the United States and analyzed all types of roads. Table 1 also shows that numerous studies have investigated the factors contributing to the severity of motorcycle crashes in the context of developing countries, such as Vietnam, Iran, India, Pakistan, and Ghana. Similarly, in Thailand, the most common modes of transportation include passenger cars and motorcycles (15 million and 21.42 million registered vehicles, respectively, in 2016). However, each country has a different traffic environment, vehicle characteristics standard, and human characteristics (e.g., culture and tradition), which may affect the crash rate and resulting injury severity differently (spatial instability). Additionally, to date, there has been no research specifically focused on the factors that affect the likelihood of death

from motorcycle-involved crashes on local roads in Thailand. By isolating the model in this way, it is possible to examine the impact of specific contributing factors on the severity of crashes [17]. The use of the random parameter logit model as an estimation parameter in the majority of models allows for the analysis of the heterogeneity of each crash [18]. If any factor is identified as a random parameter, its effect will be shown on a case-by-case basis. The addition of an unobserved heterogeneity analysis also allows for the identification of factors that increase or decrease the severity of crashes. Unobserved heterogeneity in variances reveals how factors impact the variance of a random parameter, either increasing or decreasing it. Previous studies have demonstrated that this method of data analysis provides more insight and more accurate predictions than traditional methods [18,19].

**Table 1.** Studies of Motorcycle crash severity including rider's factor.

| Authors (Year) | Research Objective/Results of Rider's Age | Compare Rider's Age | Road Type | Country (Data Year) | Parameter Estimation/Method |
|---|---|---|---|---|---|
| This Study | The study focused on examining the factors that influence the severity of injuries sustained in accidents, with a specific focus on comparing the experiences of young riders and older drivers. The authors subsequently made policy recommendations based on their findings. | Yes | Local Road | Thailand (2015–2020) | Random Parameter Logit Model with Unobserved Heterogeneity in Means and Variances |
| Kashani et al. [20] | The purpose of this research was to examine the traits of motorcycle drivers and passengers, as well as other factors that could influence the likelihood of fatalities among motorcyclists involved in traffic accidents. The result showed that older operators (>50-year-old) have a higher likelihood of fatality in a motorcycle crash. | No | Not Specific | Iran (2009–2012) | Binary Logistic Regression Model |
| Sivasankaran et al. [21] | The purpose of this research is to examine the risk factors related to single-motorcycle accidents. In regards to characteristics of the motorcyclist, it has been found that those in the younger (18–25) and working-age (25–64) groups are less likely to sustain severe injuries in single-motorcycle collisions as compared to older motorcyclists. | No | Not Specific | India (2009–2017) | Ordered Logit Model |
| Naqvi and Tiwari [22] | The goals of this paper are to examine the patterns of motorcycle crashes and determine the factors that contribute to fatal motorcycle accidents on selected highways with different lane configurations. However, age-associated factors were not considered in this study. | No | National Highways | India (2009–2013) | Binary Logit Model |
| Khan et al. [23] | The purpose of this research was to evaluate the severity and types of accidents involving motorized two-wheelers that are caused by clothing issues in Pakistan's major urban area. Additionally, it determined if females are disproportionately affected by injuries related to clothing while operating two-wheelers. Riders (older than 45-year-old) had significantly more severe injuries as compared to other age groups. | No | Not Specific | Pakistan (2007–2009) | New Injury Severity Score (NISS) computation |
| Waseem et al. [24] | This research study aimed to investigate the factors influencing motorcycle injury severity. The findings showed that the probability of severe or fatal injury increases for crashes involving middle-aged riders (25–50 years). | No | Not Specific | Pakistan | Random Parameter Approach with Heterogeneity in Means and Variances |
| Pervez et al. [25] | The goal of this study is to get better insight into the possible risk elements that contribute to the severity of the single-motorcycle collision. The results showed that a younger motorcyclist (<25) was less likely to be fatally injured and the likelihood of fatal injury decreases by 2% for younger riders compared to the older rider group. | No | Not Specific | Pakistan | Random Parameter Approach with Heterogeneity in Means and Variances |
| Nguyen et al. [26] | The study aims to identify what factors contribute to the severity of motorcycle accidents, with a specific focus on small-displacement motorcycles. The findings suggest that as the age of the driver of the first-party vehicle increases, the severity of the second-party vehicle decreases (OR = 0.963). However, when the age of the motorcycle rider involved in the crash as the first party increases, the likelihood of more severe injury also increases (OR = 1.028). | No | Not Specific | Vietnam | Ordinal Logistic Regression Model |
| Wang [27] | The objective of this study was to investigate the factors that contribute to fatalities among motorcyclists, with a particular emphasis on comparing single-motorcycle crashes to multiple-motorcycle crashes. The age of the riders are young (under 24 years old), middle-aged (25–54 years old), and old (55 years and above). The analysis revealed that young riders had a higher mortality rate compared to older riders. | No | Not Specific | Taiwan (2016 to 2020) | Mixed Logit Model |
| Wali et al. [28] | The aim of this study was to compare the anatomical injuries, including the injury severity score (ISS) and the new injury severity score (NISS), sustained in motorcycle crashes. The results of the analysis indicated that age was not a significant factor in relation to these injuries. | No | Highway | USA (2011 and 2016) | Ordered Logit Model |
| Tamakloe et al. [29] | The purpose of this study was to identify the factors that influence the severity of injuries sustained in crashes at intersections, specifically comparing signalized intersections to non-signalized intersections. The results showed that drivers under the age of 30 were more likely to suffer fatal injuries compared to those aged 30–50. | No | Not Specific | Ghana (2016 to 2018) | Binary Logit Regression Model/Association Rules Mining (ARM) |

**Table 1.** *Cont.*

| Authors (Year) | Research Objective/Results of Rider's Age | Compare Rider's Age | Road Type | Country (Data Year) | Parameter Estimation/Method |
|---|---|---|---|---|---|
| Lin et al. [30] | The study examined the influence of environmental factors on the severity of injuries sustained by motorcyclists in the vicinity of a university. The results indicated that drivers aged 45 and above were more likely to suffer severe injuries compared to those aged 25–44. | No | Not Specific | Taiwan (2017) | Logistics Regression |
| Islam [31] | The purpose of this study was to investigate the factors that influence the severity of injuries sustained by motorcyclists in work zones. The results of the analysis revealed that older riders (50–65 years) had an increased risk of death. | No | Not Specific | Florida, USA (2012–2016) | Mixed Logit with Heterogeneity in Means and Variance |
| Ijaz et al. [32] | This study examined the injury severity of various factors, dividing them into two categories: helmet-wearing and non-helmet-wearing. The principle of temporal instability was used to highlight the difference in the time dimension. | No | Not Specific | Rawalpindi, Pakistan (2017–2019) | Random Parameter Approach with Heterogeneity in Means and Variances |
| Goodwin et al. [16] | This study explored the concept of driving experience, with a focus on the age of the drivers. The models were divided into two groups: novices or young riders and returning or older riders. The results of the study showed similar trends. | Yes | Not Specific | North Carolina, USA (1991–2018) | Power Function |
| Champahom et al. [33] | The study analyzed several factors that influence the severity of injuries sustained by motorcyclists on arterial roads. The results showed that young riders (15–19 years) had an increased risk of death. | No | Arterial Road (Highway) | Thailand (2011–2017) | MCA and Ordered Logistics Regression |
| Wankie et al. [34] | This study involved interviewing commercial motorcycle riders to gather information about various dimensions, including driving experience. The results indicated that a driver's age was not a significant factor in severe crashes. However, more experienced drivers were found to be more likely to be involved in crashes. | No | Not Specific | Bamenda, Cameroon (2017) | Multivariable Logistic Regression Models |
| Pervez et al. [35] | This study examined the factors that influence the severity of motorcycle crashes. The results showed that drivers under the age of 25 and over the age of 54 were more likely to suffer fatal injuries compared to those aged between 25 and 40 years old. | No | Not Specific | Karachi, Pakistan (2014–2015) | Random Parameter Logit Model |
| Islam [10] | This study investigated the impact of the age of riders on the severity of injuries sustained in single-motorcycle crashes. Riders were classified into three age groups: under 30 years old, 30–49 years old, and 50 years old and above. The transferability test results indicated that all three models should be analyzed separately and the parameter estimation results were distinctively different. | Yes | Not Specific | Florida, USA (2016) | Mixed Logit with Heterogeneity in Means and Variance |
| Farid and Ksaibati [36] | The purpose of this study was to identify the factors that influence the severity of injuries sustained in motorcycle crashes. The two models examined were single-motorcycle crashes and multiple-vehicle crashes involving motorcycles. The analysis of the age factor revealed that younger riders involved in a single-motorcycle crash had a lower risk of death compared to middle-aged riders (30–59 years). | No | Not Specific | Wyoming, USA (2008–2017) | Mixed Binary Logistic Regression Model |
| Abrari Vajari et al. [37] | This study analyzed the factors that influence the severity of injuries sustained in motorcycle crashes at intersections. The results showed that riders aged over 59 years were less likely to be injured compared to riders of other ages. | No | Not Specific | Victoria, Australia (2006-2018) | Multinomial Logit Model |
| Lam et al. [14] | This study examined the injury severity of light motorcycle crashes (50 and 250 cc) by interviewing crash victims. The results showed that drivers aged over 65 were more likely to suffer serious injuries. | No | Not Specific | Taiwan (2015–2017) | Binary Logistic Model |
| Hidalgo-Fuentes and Sospedra-Baeza [38] | This study used a large dataset of crash data to investigate the distribution of crash occurrences among motorcyclists by gender and age group. The results showed that female motorcyclists aged 16–34 years old and 55–64 years old had a high risk of fatality, while male motorcyclists in the age range of 65 years and above were at increased risk based on a statistical analysis of the factors that influence injury severity. | No | Not Specific | Spain (2006–2011) | Descriptive Statistics (Custom Table and Graph) |
| Alnawmasi et al. [15] | This study applied the principle of temporal instability analysis. The results showed that riders older than 60 had an increased risk of severe injury. | No | Not Specific | Florida, USA (2005–2015) | Random Parameter Logit Model with Unobserved Heterogeneity in Means and Variances |
| Wali et al. [39] | This study examined the risk factors for motorcycle injury crashes. While the study did not specifically address the age of riders, it did find that increased driving experience was associated with a reduced risk of crash severity. | No | Not Specific | California, USA | Random Parameter Logit Model with Heterogeneity-in-Means |

Statistical data from Figure 2 indicate that the number of motorcycle crashes is more common among drivers under the age of 30, which may be due to their tendency to engage in faster and more aggressive driving behaviors [15]. However, when examining the severity of these crashes, the data show that young riders had more deaths during the years 2015–2017, while older riders had a higher number of deaths during 2018–2019 and 2020. Additionally, the fatalities–per–crash ratio suggests that older drivers are more likely to die in motorcycle crashes than younger drivers.

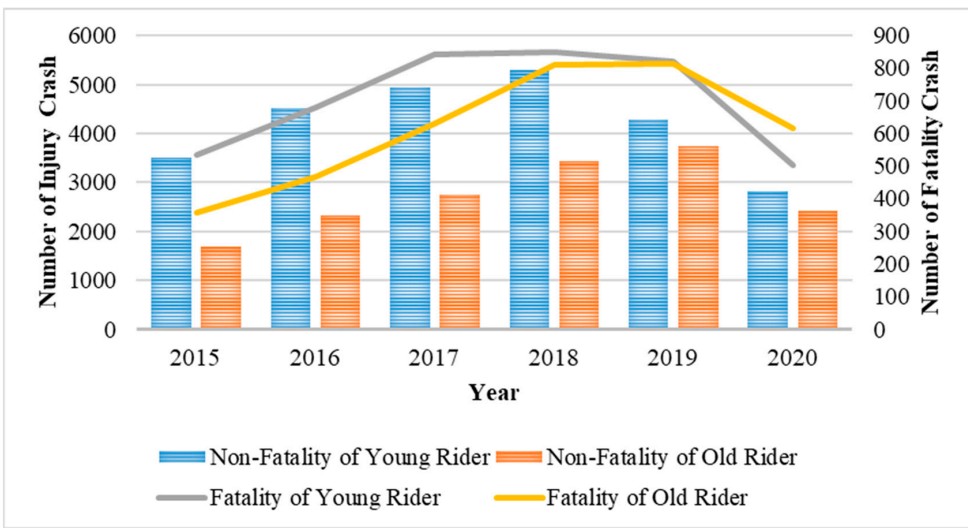

**Figure 2.** Crash severity of 2015–2020 (Comparison of riders' age).

It is clear from the literature review that there is a gap in research on the comparison of motorcycle crash severity on local roads. This study aims to address this gap, despite the fact that local roads account for the second highest number of deaths from motorcycle crashes, even though the number of fatalities is generally lower on these roads compared to highways. Most research has primarily focused on crashes on highways, leaving the issue of motorcycle crashes on local roads largely unexamined. This research seeks to study motorcycle crashes on local roads in order to identify appropriate guidelines or measures to reduce the death rate from these crashes. By separating the models, it is possible to gain a more detailed understanding of the factors contributing to crash severity, which can inform policy recommendations for road safety agencies, such as local government administrations and the Department of Transport. In terms of implementation, access-oriented roads are a priority in Thailand and other countries with similar traffic environments characterized by a high proportion of motorcycles.

This research paper is organized into several sections. The Section 1 presents the statement of the research problem. The Section 2 reviews the factors that affect motorcycle crash severity and the potential differences in the effects of these factors between young and old riders. The Section 3 describes the data used for analysis and model development. The Section 4 presents the results of parameter estimation and the testing of the suitability of the two models for separate analysis. The Section 5 discusses the results of the parameter estimation, and the Section 6 provides a conclusion of the findings and recommends policies based on the parameter estimation results.

## 2. Contributing Factors

This section aims to examine the literature on the factors that have been identified as potentially influencing the likelihood of fatal outcomes in motorcycle crashes. In particular, we will review the research on the role of the driver's age, as well as other variables that have been examined in previous studies. Our review reveals that these variables can be broadly grouped into 4 categories, comprising a total of 22 variables. These categories and the corresponding variables are described in detail below.

### 2.1. Rider Characteristics

Research has identified several driver characteristics that may affect driving behavior and the use of safety equipment. One such variable is gender, with several studies finding that male drivers tend to engage in more risky behaviors, such as speeding and overtaking, compared to female drivers [38]. Another variable of interest is the nationality of the driver, as foreign drivers may be unfamiliar with local traffic laws and routes, which

could increase the risk of collision [40]. Additionally, the familiarity of the driver with the crash location, as reflected in whether the driver resides in the same district as the crash scene, may also play a role in the likelihood of a fatal outcome [40]. The use of safety equipment, such as helmets, can significantly reduce the risk of death in a motor vehicle accident [32], although the protective effect of helmets may vary depending on the age of the driver [32]. The use of alcohol or other intoxicants by the driver has also been identified as a factor that may affect the severity of a crash, with some studies finding that intoxication increases the risk of death [30], while others have found a lower risk of fatal outcomes in crashes involving intoxicated drivers [41]. Holding a valid driver's license may also be associated with a lower risk of fatality in a motor vehicle accident, as it suggests that the driver is knowledgeable about traffic laws and has received some form of training in driving skills [39,42]. However, the relationship between age and crash outcomes may be more complex, with some research suggesting that younger drivers may have an advantage in terms of their ability to react quickly and make decisions in challenging driving situations [15].

*2.2. Crash Characteristics*

This group of variables pertains to the hypothesized causes of the collision and their relationship to crash severity in single- and multi-vehicle accidents. It is well established that multi-vehicle crashes are more likely to result in fatalities, particularly when they involve motorcycles and larger vehicles [43]. However, these studies have largely focused on major roads and mobility, rather than accessibility [33]. The "rider at-fault" variable refers to situations where the driver is the cause of the crash. Exceeding the speed limit is another factor that has been consistently linked to increased risk of death in motorcycle accidents [44]. Traffic violations, such as running a red light or driving in the wrong direction, also increase the likelihood of fatal outcomes due to the higher impact forces involved in these types of collisions [45]. Illegal overtaking, which refers to passing another vehicle in a prohibited area marked by solid traffic lines, can also lead to head-on collisions and a higher risk of death [46]. The use of mobile phones while driving has been identified as a potentially hazardous behavior, although it is more commonly associated with other types of vehicles [44]. Drowsy or asleep driving is another factor that may be more prevalent in other vehicle types, but it can also pose a risk to motorcycle riders [44]. Finally, the "close cut-in-front" variable refers to a collision that is caused by another vehicle cutting in front of the motorcycle, which is more likely to affect the frequency of crashes rather than their severity [47].

*2.3. Environmental Characteristics*

The environmental variable in this study focuses on the impact of weather conditions on driving visibility, specifically the effect of rain on crash outcomes. Previous research has found that driving in rainy conditions tends to result in slower speeds, which can reduce the severity of crashes and decrease the risk of fatalities [41]. Similarly, driving in fog or dust may also reduce visibility and encourage caution, leading to a lower risk of fatalities in the event of a crash [41]. These effects may be similar across age groups. Nighttime crashes, on the other hand, have been found to be associated with an increased risk of death in motorcycle accidents, particularly when the road is open and the lighting is poor [48]. This may be due to the fact that brighter lighting conditions can boost riders' confidence and encourage higher speeds, which can lead to more severe impacts in the event of a collision. Finally, crashes that occur at night with no lights present have also been linked to an increased risk of fatal outcomes, as drivers may not have sufficient time to react to oncoming traffic or obstacles in the roadway [31].

*2.4. Road Characteristics*

Among the variables in this group, road characteristics may also play a role in determining the likelihood of fatal outcomes in motorcycle crashes. In this study, which focuses

on local roads, there may be variations in the quality and type of road surface that could affect the severity of injuries sustained in a crash. For example, local roads may not be built to withstand heavy traffic and may be more prone to damage compared to highways or rural roads. Wet roads may also increase the risk of crashes due to slippery conditions, although the severity of these accidents may be less severe due to drivers being more cautious and reducing their speeds [49]. Road surfaces that are under construction or in poor repair may also pose an increased risk of crashes due to the lack of proper warning signs or inadequate measures to prevent non-standard construction [31]. The roughness or smoothness of the road surface may also be a factor, with poorly maintained roads being more prone to causing accidents [31]. The curvature of the road has also been identified as a significant factor in crash outcomes, with studies finding that motorcycle accidents on curves have a higher risk of death compared to straight stretches of road [50]. This may be particularly true for young drivers, who may be more prone to engaging in risky behaviors, such as speeding around curves [50].

## 3. Method

### 3.1. Data and Empirical Setting

The data for this study were collected by the Department of Disaster Prevention and Mitigation in the Ministry of Public Health. Local authorities conducted surveys on each crash case, gathering information, such as the type of road the crash occurred on (highway, rural road, or local road), the physical characteristics of the crash site (such as the presence of curves), the types of vehicles involved, details about the victims (such as gender and age), and potential causes of the crash. This study focused specifically on motorcycle crashes that occurred on local roads from 2015 to 2020.

To prepare the data for analysis using a logit model, the dependent variable was coded as either 0 (if the crash resulted in injuries) or 1 (if it was fatal). Similarly, contributing factors were also coded as either 0 or 1; for example, if the driver was male, the value would be set to 1, while it would be set to 0 for a female driver. The specifics of both models are presented in Table 2.

**Table 2.** Descriptive Statistics.

| Variable | Description | Young Rider | | Old Rider | |
|---|---|---|---|---|---|
| | | Mean | S.D. | Mean | S.D. |
| Fatal Crash (Fatal/Non-fatal crash) | 1 = Fatal; 0 = Injury | 0.143 | 0.350 | 0.184 | 0.387 |
| *Rider characteristics* | | | | | |
| Gender | 1 = Male; 0 = Female | 0.756 | 0.430 | 0.726 | 0.446 |
| Foreigner | 1 = Foreigner; 0 = Thai | 0.025 | 0.155 | 0.012 | 0.109 |
| Local address | 1 = Local address; 0 = Otherwise | 0.713 | 0.452 | 0.824 | 0.381 |
| Helmet | 1 = Wearing helmet; 0 = Otherwise | 0.557 | 0.497 | 0.558 | 0.497 |
| DUI | 1 = Under influence of Alcohol; 0 = Otherwise | 0.152 | 0.359 | 0.165 | 0.371 |
| License status | 1 = Unlicensed rider; 0 = License rider | 0.939 | 0.240 | 0.976 | 0.153 |
| *Crash characteristics* | | | | | |
| Multi-Vehicle Crash | 1 = Two-vehicle crash; 0 = Single-crash | 0.054 | 0.327 | 0.035 | 0.272 |
| Rider at fault | 1 = At fault; 0 = Not at fault | 0.766 | 0.423 | 0.773 | 0.419 |
| Exceeding PSL | 1 = Exceeding the speed limit; 0 = Otherwise | 0.336 | 0.472 | 0.277 | 0.447 |
| Violation | 1 = Involved traffic sign/signal/wrong-direction violation; 0 = Otherwise | 0.005 | 0.069 | 0.005 | 0.070 |
| Overtaking | 1 = Illegal/improper overtaking; 0 = Otherwise | 0.015 | 0.120 | 0.012 | 0.109 |
| Mobile use | 1 = Using Mobile phone; 0 = Otherwise | 0.004 | 0.060 | 0.002 | 0.050 |
| Asleep | 1 = Asleep/fatigue; 0 = Otherwise | 0.004 | 0.066 | 0.006 | 0.077 |
| Cutting in front | 1 = Hitting object cutting in front; 0 = Otherwise | 0.171 | 0.377 | 0.163 | 0.369 |
| *Environmental factor* | | | | | |
| Rain | 1 = Under rainy weather; 0 = Otherwise | 0.028 | 0.164 | 0.025 | 0.157 |
| Dust/foggy | 1 = Under dust/foggy weather; 0 = Otherwise | 0.030 | 0.172 | 0.023 | 0.149 |
| Lit road | 1 = Nighttime and lit road; 0 = Daytime | 0.219 | 0.414 | 0.134 | 0.341 |
| Unlit road | 1 = Nighttime and unlit road; 0 = Daytime | 0.191 | 0.393 | 0.140 | 0.347 |
| *Road characteristics* | | | | | |
| Wet road | 1 = Wet road surface; 0 = Otherwise | 0.035 | 0.185 | 0.030 | 0.171 |
| Work zone | 1 = Road construction work zone area; 0 = Otherwise | 0.001 | 0.033 | 0.001 | 0.037 |
| Road surface | 1 = Broken road surface; 0 = Good road surface | 0.026 | 0.160 | 0.032 | 0.176 |
| Alignment | 1 = Horizontal curve; 0 = Straight road | 0.131 | 0.338 | 0.120 | 0.325 |

Note: N of young rider = 29,610 and N of old rider = 20,079.

Analysis of the crash data showed that older drivers had a higher fatality rate compared to young drivers. Examination of the contributing factors, specifically the driver factor group, revealed that a majority of them were male. There was a notable difference in the proportions of local victims, with older riders being more represented than young riders. For crash characteristics, the overall proportions for both age groups were similar in direction, but there were significant differences in the proportions of certain factors, such as exceeding the speed limit. It appears that young drivers were more likely to engage in risky behaviors, such as speeding, as evidenced by the higher proportion of young drivers compared to older drivers for this factor. The environmental factor of nighttime crashes (with or without lights) was found to be significantly more prevalent among young riders than older riders. This suggests that young riders may be more prone to engaging in risky behaviors, possibly due to a tendency towards more aggressive driving.

### 3.2. Parameter Estimation

This study aims to investigate the relationships between independent variables (contributing factors) such as gender, helmet usage, possession of a valid driver's license, and involvement in a crash during rainy conditions, and the dependent variable of motorcycle crash severity, which was classified as either fatal or non-fatal. Data analysis was conducted using statistical modeling techniques that incorporated the concept of a random parameter, which refers to a variable that exhibits random variation. If a factor is found to be a random parameter (as indicated by a significant standard deviation), it suggests that such a factor may influence crash severity and may increase or decrease the likelihood of death. The distribution of standard deviations follows a normal distribution form [51]. The unobservable heterogeneity of each motorcycle crash is described in terms of heterogeneity in mean and heterogeneity in variance. Model development involves specifying the values of the random parameters [52].

$$S_{jm} = \beta_j X_{jm} + \varepsilon_{jm}, \tag{1}$$

where $S_{jm}$ is the motorcycle crash injury at $j$ level in a motorcycle crash. $\beta_j$ denotes the relationship between the independent variable $X_{jm}$ (e.g., wearing a helmet, exceeding speed limit, crashing on an unlit road, etc.) affecting fatalities in motorcycle crashes, and $\varepsilon_{jm}$ refers to the occurring standard error. The next step was to identify the unobserved heterogeneity details potentially resulting in the probability of the dependent variables obtained from random-parameter evaluation.

$$P_m(j) = \int \frac{EXP(\beta_j X_{jm})}{\sum_{\forall j} EXP(\beta_j X_{jm})} f(\beta|\rho) d\beta, \tag{2}$$

where $P_m(j)$ is the probability of injury at $j$ level in a motorcycle crash, and $f(\beta|\rho)$ represents the density function of $\beta$ together with $\rho$. This is a parameter value that shows the mean and variance in order to identify the probability of unobserved heterogeneity in the means and variances of random parameters. Thus, $\beta_{jm}$ was allowed to vary with each motorcycle crash occurrence.

$$\beta_{jm} = \beta_j + \Theta_{jm} Z_{jm} + \sigma_{jm} EXP(\omega_{jm} W_{jm}) \nu_{jm}, \tag{3}$$

where $\beta_j$ represents the mean of parameter estimation on the basis that each motorcycle crash is different. $Z_{jm}$ denotes the relationship between the independent and dependent variables, with an attempt to identify unobserved values that will affect the mean of the parameter estimates. $\Theta_{jm}$ shows the properties of unobserved heterogeneity that affect the mean of the slope. $W_{jm}$ refers to the effect of unobserved heterogeneity affecting the variance $\sigma_{jm}$ of the slope parameter estimation, while $\nu_{jm}$ is the disturbance term.

The selection of the most suitable model was determined based on the approximation of the empirical data using values [53]

$$\rho^2 = 1 - \frac{LL(\beta)}{LL(0)} \tag{4}$$

where $LL(\beta)$ is a log-likelihood of the model with parameter estimation, and $LL(0)$ is the log-likelihood of the model with parameter estimation with constant only [54]. The suitable value is not clearly determined. However, road crash models can be acceptable with values greater than 0.1 [17,18]. However, Yan et al. [55] accepted models with $\rho^2$ greater than 0.06.

For all collision cases, the average marginal effect (ME), which is used to consider the effect of factors on crash severity, is taken into account. ME is a value that indicates how the interpretation of changing one unit of a variable (in this study, a change from 0 to 1), such as from a Thai rider to a foreign rider in a motorcycle crash on local roads, affects how much the probability of deaths from motorcycle crashes on local roads changes [15].

Nlogit version 6 was used to develop a random parameter logit model with unobserved heterogeneity in means and variance. A Halton sequence of draws was used to develop the model because there are speed gains with no degradation in simulation performance [56]. The model in this study was set at 500 Halton [31].

*3.3. Transferability Test*

The transferability test determines whether the motorcycle crash dataset should be separately analyzed on the basis of likelihood ratio tests, which are used to compare two individual riders' ages [57], as follows:

$$\chi^2 = -2\left[ LL\left(\beta_{Young\ and\ old\ rider}\right) - LL\left(\beta_{young\ rider}\right) - LL(\beta_{old\ rider})\right] \tag{5}$$

where $\beta_{Young\ and\ old\ rider}$ is the convergence of the model that used all of the available data (in this study, motorcycle crashes on local roads involving riders younger than 30 years and those older than 50 years), and $\beta_{young\ rider}$ is the convergence of the model that used data of young riders. $\beta_{old\ rider}$ is the convergence of the model that used the data of older riders. For degree of freedom (df), it is the number of estimated parameters in each model. The null hypothesis is that both sets of data have no transferability, implying that there is no need to separate the models. The analysis results show that with $\chi^2$ = 492.098 and df = 22, it will obtain 99.99%, so the null hypothesis was rejected. This indicates that crash data for young and old rider models should be analyzed separately [58].

## 4. Results

*4.1. Descriptive Statistics*

The mean value for fatal crashes among young riders was lower than that of older riders, indicating that older riders were more likely to succumb to their injuries in a crash. Most of the variables in the driver characteristics group had similar means. One notable exception was gender, with males having more crashes than females, and the helmet wear rate being approximately 55%. Another variable that differed significantly was foreign drivers, with a higher proportion of old riders represented. For crash characteristics, variables such as rider at fault and cutting in front were similar between the two age groups. However, exceeding the speed limit was found to be more prevalent among young riders. In terms of environmental variables, crashes occurring in rain, smoke, and fog had similar proportions for both age groups. Nighttime crashes, with or without lights, had a higher proportion of young riders. Most road characteristics, such as wet roads and rough road surfaces, did not show significant differences between the two age groups. However, young riders were more likely to be involved in crashes on curves, while older riders were more likely to crash on straight roads.

### 4.2. Model Results

The results of the random parameter logit model with unobserved heterogeneity in means and variances for the young and old models, respectively, had goodness-of-fit values of 0.0870 and 0.0757, which were relatively small, but within the acceptable range [55]. The Young model (Table 3) had several significant variables, including three random parameters: local address, rain, and road surface. The variables identified as having unobserved heterogeneity consisted of two variables: violation and overtaking. Significant unobserved heterogeneity in means was found in two pairs, namely, rain: overtaking and road surface: overtaking. Significant unobserved heterogeneity in variances was found in only one pair, i.e., rain: violation.

**Table 3.** Young Rider Model Result.

| Variable | Parameter Estimation | | Marginal Effect |
|---|---|---|---|
| | Coefficient | t-Stat | |
| Constant | −1.852 ** | −37.97 | |
| Rider characteristics | | | |
| Gender | 0.645 ** | 17.61 | 0.10552 |
| Foreigner | 0.137 * | 1.78 | 0.02249 |
| Local address | −0.384 ** | −13.35 | −0.06277 |
| S.D. Local | 0.916 ** | 36.97 | |
| Helmet | −0.112 ** | −4.19 | −0.01833 |
| DUI | −0.464 ** | −11.19 | −0.07595 |
| License status | −0.759 ** | −10.80 | −0.12425 |
| Crash characteristics | | | |
| Multi-Vehicle Crash | −0.227 ** | −2.63 | −0.03722 |
| Rider at-fault | −0.199 ** | −6.63 | −0.03261 |
| Exceeding PSL | 0.827 ** | 31.39 | 0.13536 |
| Mobile use | −0.589* | −1.81 | −0.09633 |
| Asleep | 0.377 ** | 2.19 | 0.06166 |
| Cutting-in-front | −0.215 ** | −5.58 | −0.03523 |
| Environmental characteristics | | | |
| Rain | −0.657 ** | −3.87 | −0.10759 |
| S.D. of Rain | 2.119 ** | 11.58 | |
| Dust/foggy | −0.432 ** | −4.89 | −0.07073 |
| Lit road | 0.352 ** | 11.16 | 0.05764 |
| Unlit road | 0.467 ** | 14.10 | 0.07636 |
| Road characteristics | | | |
| Wet road | -0.131 | −1.10 | −0.02146 |
| Work zone | 0.677 ** | 2.22 | 0.11081 |
| Road surface | −0.831 ** | −5.41 | −0.13596 |
| S.D of Road Surface | 2.063 ** | 10.72 | |
| Alignment | 0.104 ** | 2.81 | 0.01699 |
| Unobserved Heterogeneity in means | | | |
| Local: Overtaking | 0.021 | 0.15 | |
| Rain: Overtaking | 1.827 ** | 2.51 | |
| Road surface: Overtaking | 2.205 ** | 2.55 | |
| Unobserved Heterogeneity in variances | | | |
| Local: Violation | 0.188 | 1.29 | |
| Rain: Violation | 1.805 ** | 2.54 | |
| Road surface: Violatio | −0.027 | −0.05 | |
| Model statical | | | |
| LL(B) | −11083.572 | | |
| LL(0) | −12139.960 | | |
| McFadden $\rho^2$ | 0.0870 | | |

Note: * *p*-value < 0.1, ** *p*-value < 0.05.

The Older model (Table 4) also had several significant variables, including three random parameters: local address, helmet, and violation. The variables identified as having unobserved heterogeneity consisted of three variables: cutting-in-front, work zone, and alignment. Significant unobserved heterogeneity in means was found in four pairs, namely, local address: cutting-in-front, helmet: cutting-in-front, helmet: alignment, and violation: alignment. Significant unobserved heterogeneity in variances was found in one pair, i.e., helmet: work zone.

**Table 4.** Old Rider Model Result.

| Variable | Parameter Estimation | | Marginal Effect |
|---|---|---|---|
| | Coefficient | t-Stat | |
| Constant | −1.432 *** | −25.00 | |
| Rider characteristics | | | |
| Gender | 0.622 ** | 16.52 | 0.06758 |
| Foreigner | 0.188 | 1.51 | 0.02041 |
| Local address | −0.183 ** | −4.31 | −0.01991 |
| S.D. Local | 0.482 ** | 20.16 | |
| Helmet | −0.718 ** | −18.35 | −0.07791 |
| S.D. Helmet | 1.601 ** | 40.97 | |
| DUI | −0.734 ** | −15.11 | −0.07969 |
| License status | −0.771 ** | −5.62 | −0.08377 |
| Crash characteristics | | | |
| Multi-Vehicle Crash | −0.446 ** | −3.56 | −0.04838 |
| Rider at-fault | −0.134 ** | −3.84 | −0.01458 |
| Exceeding PSL | 0.915 ** | 28.55 | 0.09935 |
| Violation | 0.093 | 0.56 | 0.01014 |
| S.D. Violation | 1.747 ** | 6.92 | |
| Overtaking | 0.314 ** | 2.32 | 0.03407 |
| Mobile use | −1.152 ** | −2.12 | −0.12506 |
| Asleep | 0.692 ** | 4.09 | 0.07512 |
| Environmental characteristics | | | |
| Rain | 0.278 * | 1.78 | 0.03014 |
| Dust/foggy | −0.308 ** | -2.84 | −0.03344 |
| Lit road | 0.264 ** | 6.02 | 0.02865 |
| Unlit road | 0.389 ** | 9.16 | 0.04226 |
| Road characteristics | | | |
| Wet road | −0.408 ** | −2.69 | −0.04435 |
| Work zone | | | |
| Road surface | −0.144 ** | −1.57 | −0.01558 |
| Unobserved Heterogeneity in means | | | |
| Local address: Cutting-in-front | 0.121 * | 1.83 | |
| Local address: Alignment | −0.083 | −0.73 | |
| Helmet: Cutting-in-front | −0.232 ** | −3.13 | |
| Helmet: Alignment | 0.412 ** | 4.44 | |
| Violation: Cutting-in-front | −0.394 | −0.99 | |
| Violation: Alignment | −0.907 ** | −1.99 | |
| Unobserved Heterogeneity in variances | | | |
| Local address: Work zone | 0.609 | 0.92 | |
| Helmet: Work zone | 4.317 ** | 10.75 | |
| Model statical | | | |
| LL(B) | −8859.72489 | | |
| LL(0) | −9585.0163 | | |
| McFadden $\rho^2$ | 0.0757 | | |

Note: * *p*-value < 0.1, ** *p*-value < 0.05. *** *p*-value < 0.01.

## 5. Discussion

### 5.1. Young Rider Model

The rider characteristics were found to be significant predictors of fatal crashes. Gender was found to be a significant predictor, with males being more likely to be involved in fatal crashes. This may be due to the fact that males tend to drive faster than females, particularly on local roads with less traffic. Similarly, although the ages of riders were not reported, a study in Vietnam also found that male gender increased the odds of more severe injuries by almost four times [26]. Foreigner status was also found to be a significant predictor, indicating that foreign riders are more likely to be involved in fatal crashes. This may be due to unfamiliarity with the route and traffic rules in Thailand.

Local address was found to be a significant predictor, with victims who live in the same district as the crash scene being less likely to be involved in a fatal crash. This may be because young riders who are familiar with frequent crash points are able to be more careful than outsiders [40]. However, this variable was found to be a random parameter, with approximately 33.7% of this factor having a high probability of death (Figure 3(A1)). This may be due to the fact that, even though some groups of young riders are familiar with the local area, their careless or aggressive driving behavior still puts them at risk of death (this random parameter may have potentially uncovered the effect of an unobserved characteristic that indirectly affects the outcome severity through the risk-compensating behavior of young familiar riders). Helmets were found to have a significantly negative impact, suggesting that wearing a helmet can reduce the likelihood of death. This finding is consistent with numerous studies [59,60] and research from developing countries [20,22,35] that have confirmed the protective effect of helmet use.

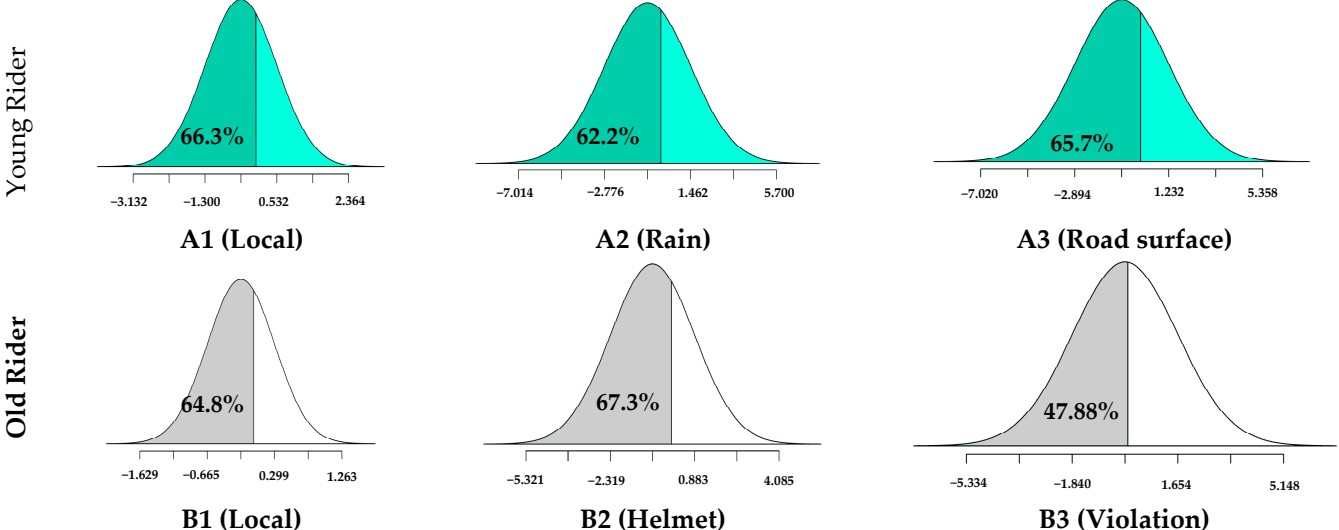

**Figure 3.** Scale parameters for distribution of the contributing factor.

Drunk driving was found to have a negative association with fatal crashes. This may be because the riders (mostly local people) in the majority of drunk-driving cases ride their motorcycles home on a short-distance trip on local roads, using low-speed driving. This finding is similar to past studies in developing countries, which found that intoxicated victims were less likely to be killed in motorcycle crashes in a 2019 dataset [40]. However, recent motorcycle crash research in Pakistan found that crashes involving motorcyclists under the influence of drugs or alcohol increased the likelihood of fatal injury by 2% [25]. This indicates a significant variation that may be attributed to the difference in riders' characteristics and their attitudes/behaviors, which may have influenced the culture and traditions in different countries.

Another significant predictor was having a driver's license. This was found to be a protective factor, with licensed riders being less likely to be involved in fatal crashes. This

finding is reasonable, as those who have passed training and driving tests are likely to have more driving skills than those without a driver's license, allowing them to make better decisions to brake in time and reduce the likelihood of death. This finding is consistent with the findings of Wang [27], who found that unlicensed riders were more likely to be killed in motorcycle single-vehicle crashes. Finding of previous research in Vietnam and Pakistan also suggest the safety benefit of obtaining a license for motorcycle riders [26,35].

The crash characteristics were found to be significant predictors of the severity of the crash. Multi-vehicle crashes were found to be less severe, with one possible explanation being that motorcycles are commonly found on local roads, and when they collide with each other, the impact is not as severe. In contrast, single crashes where motorcycles tend to drive fast and collide with fixed objects, such as trees or lampposts, were found to be more severe. This finding is consistent with a previous study using data from Pakistan, which found that collisions with fixed objects on motorcycles increased the likelihood of fatalities [25]. Another study from Ghana also found that multi-vehicle crashes in locations that do not encourage speeding (e.g., local road, intersection, or congested area) are less likely to result in death and serious injury [29].

The rider being at fault was found to be a protective factor, with this variable being negatively correlated with crash severity. This may be because in cases where the rider is at fault, they tend to make a cut in front, which can reduce the impact force in a collision with other vehicles that are not driving at high speeds. This finding is consistent with the finding of the previous study in Vietnam that reported that second-party (non-at-fault party) motorcycle crashes were likely to result in less severe consequences than first-party (at-fault party) crashes [26].

Exceeding the speed limit was found to be a significant predictor of crash severity, with young drivers who drive at high speeds being more likely to be involved in severe crashes. This finding confirms that even on local roads, high speeds can lead to a high risk of death [61], which is also in line with numerous studies from developing countries [25,29,35]. Using a cellphone while driving was found to have a mitigating effect on crash severity, with drivers who were using their phones while driving being less likely to be involved in severe crashes. This could be due to the fact that drivers who are using their phones are more mindful of their safety, and as a result, they do not drive at high speeds. Falling asleep while driving was found to be a significant predictor of crash severity, with drivers who fell asleep being more likely to be involved in severe crashes, which is consistent with a finding of past research in Vietnam [44]. This may be because drivers who fall asleep while driving are unable to reduce their speed in time, leading to a high risk of death [51]. In terms of unobserved heterogeneity, violations and overtaking were found to not be significant predictors of crash severity. This may be because violations, such as overtaking where it is prohibited or disobedience against traffic lights, are rare.

The environmental factors were found to be significant predictors of fatal crashes. Based on the average marginal effect, crashes that occurred in rainy weather were found to be less likely to result in fatalities. It is noteworthy that this variable was also found to be a random parameter, with a 62.2% of such crashes having a low chance of death versus 37.8% having a high chance of death (Figure 3(A2)). Collisions that occurred in dusty or foggy conditions were found to have a low chance of death. This may be due to the fact that rain, fog, or smoke can affect visibility and cause drivers to be more cautious, as found in the study by Abrari Vajari et al. [37]. Several previous studies from developing countries also found that motorcycle crashes occurring during clear/dry weather increased the likelihood of severe and fatal injuries, compared to adverse weather conditions [24,29].

Crashes that occurred at night were found to be more likely to result in fatalities, with both lit and unlit roads associated with a higher chance of death. This may be because at night, the openness of local roads (low traffic volume) allows young drivers to drive at high speeds. Additionally, crashes on unlit roads were found to be more severe than crashes on lit roads. This finding is reasonable, as lit roads may allow riders to notice objects and reduce their speed before a collision, as suggested by [62]. Tamakloe et al. [29]

and Pervez et al. [35] also reported similar findings using crash data from Ghana and Pakistan, respectively.

Road characteristics were found to be significant predictors of fatal crashes. Crashes that occurred on roads under construction were found to have a high chance of resulting in death. The cause of this issue could be attributed to the use of non-conventional signage or safety measures on roads undergoing construction or maintenance, which, combined with the reckless driving habits of teenagers, can lead to collisions with equipment or barriers in the work zone due to riders' inability to slow down in time. Crashes on rough roads were found to be a random parameter, with approximately 65.7% of cases having a low chance of death, and the remaining 34.3% having a high chance of death (Figure 3(A3)). It may be possible that some groups of young drivers drive cautiously, particularly when they are familiar with the riding conditions they are in (e.g., rough roads). When accidents happen, these drivers tend to lose control and crash on their own, rather than crashing into stationary objects at high speeds. Additionally, a smaller group of riders who are at a higher risk of fatal injury may consist of inexperienced or unfamiliar young riders who are not aware of poor road conditions and ride at a normal speed.

The curve variable was found to be a significant predictor, with crashes on curves being associated with a higher chance of death. This finding suggests that when young riders drive around corner at high speeds, they may lose control and collide with the side guardrails, which could be catastrophic for motorcycle riders [48]. Similarly, a previous study in Iran also reported that the risk of death in motorcycle accidents on curves is 35% greater than those on straight sections [20].

Unobserved heterogeneity in means was found to be a significant predictor of crash severity. Specifically, the "Rain: Overtaking" variable was found to be positively significant. That is, crashes due to illegal overtaking while also occurring during raining increase the probability of riders being fatally injured. Overtaking, which involves increasing one's speed, can be dangerous in rainy conditions with limited visibility. This is because it can lead to loss of control and a potential head-on collision with another vehicle, resulting in a higher impact force and an increased risk of death. Similarly, the "Road Surface: Overtaking" variable was also found to have a positive correlation with a higher risk of death in crashes that involve illegal overtaking on rough road surfaces. It is logical that illegal overtaking on rough road surfaces would lead to a higher risk of death. This is because accelerating on a rough road can cause a loss of control and collisions with objects on the roadside or opposing vehicles, which are likely to increase the risk of death.

Unobserved heterogeneity in variances was also found to be a significant predictor. That is, the crashes involving traffic sign/signal/wrong-direction violation increase the variance of crashes occurring during raining. Such increases in variances of the random parameter make its distribution wider and increases its randomness. In simple terms, the "Violation" makes the parameter estimate of each crash observation of the random parameter "Rain" even more varied or random across crash populations, compared to the degree of randomness generated by the random parameters without considering the variances heterogeneity [63]. This, to some degree, captured the effect of unmeasured factors or characteristics related to the detailed information on variables reflecting riders' violations, which may go beyond police report capabilities, thus making the accuracy of the crash modelling more reliable [64–66].

### 5.2. Old Rider Model

Similar to the young riders, the old male riders were found to be positively associated with higher injury severity in crashes, compared to old female riders. Again, this may be due to the fact that male riders are more likely to use a higher operating speed than female riders. On the other hand, the foreign rider factor was not significant in the old rider model.

The location of the victim's residence also had an effect on the outcome of the crash. Drivers who lived in the same area as the crash scene were less likely to die. This factor also led to a significant random parameter, with 64.8% of crashes being less severe and 35.2%

being more severe (as shown in Figure 3(B1)). Although most of these crashes had a lower risk of death, the 35.2% of these crashes (that have a higher probability of fatality) may be represented by a small group of older riders who are more likely to be over 60 years old and may have weaker bodies, which increases the risk of injury severity and death [67].

Consistent with the young rider model, the use of a helmet was found to reduce the risk of fatal injury for old drivers. However, this variable was also found to be a random parameter, with 67.3% having a low chance of fatality and 32.7% having a high risk of death (Figure 3(B2)). This may be due to the use of cheap helmets (non-standard helmets with relatively cheap price) that provide less protection [68], or the fact that some older riders may have weak physical health after the crash [13]. Crashes involving intoxicated drivers and drivers with more driving experience had a lower probability of resulting in fatal injury. This suggests that older riders, if trained, can reduce the risk of death through good driving skills [40,43].

The multi-vehicle collision factor was found to have a lower chance of fatalities in this study, which is inconsistent with previous research. Abdul Manan et al. [48] found that multi-vehicle crashes were more likely to result in fatalities, whereas Zulkipli et al. [59] found that single crashes had a higher risk of injury than multiple-vehicle crashes, and Islam [31] found that single crashes were more likely to cause injury than multi-vehicle crashes in work zones. This study, however, only looked at local roads, which may explain the discrepancy with other studies. In the context of developing countries, although age and road types were not reported, previous studies also reported consistent results with this finding [25,29].

The rider at fault was found to be less likely to die in a crash, whereas exceeding the speed limit was associated with a higher chance of mortality, which is in line with previous literature [59]. The violation variable was found to be insignificant in the model, but had a significant scale as a random parameter (Figure 3(B3)), indicating that most violations had no correlation with severity, but in some cases, such as head-on collisions caused by driving in the wrong direction and overtaking where prohibited, increased the risk of death for elderly riders [36]. The use of a mobile phone while riding was negatively significant, indicating a low chance of death. Falling asleep while riding was found to have a high chance of causing death for elderly drivers, possibly due to their inability to reduce speed in a crash. The explanation for these two variables could be reasonably explained, similar to previous explanations for the young riders model.

Crashes that occurred in rainy weather were found to have a higher chance of causing death for elderly drivers, which is possibly due to their insufficient physical strength and the challenges (e.g., visibility) in providing timely assistance due to the weather conditions. In contrast, collisions that occurred in dusty or foggy weather were found to have a lower chance of causing mortality, which is also in line with past research [69]. Crashes that occurred at night, on both lit and unlit roads, were found to have a higher chance of causing death [37]. These findings suggest that weather and lighting conditions can have a significant impact on the severity of a crash and the likelihood of death for elderly riders. Although the ages of riders were not reported, past studies also reported similar findings in the context of developing countries [29,35].

Two road characteristics were found to be significant in this study. Wet roads were found to decrease the chance of fatalities, which makes sense, as slippery conditions can cause riders to take safety precautions and slow down, resulting in less severe accidents [55]. A study in Iran [20] also reported that motorcycle accidents tend to be less severe on wet roads, particularly during winter months. Crashes on rough road surfaces were also found to have a lower chance of causing death for elderly drivers, possibly because they are more cautious and ride more slowly in anticipation of potential hazards.

In terms of unobserved heterogeneity in means, the variable "Cutting-in-front" increases the mean of "Local address", thus making the risk of death more likely for old riders. In contrast, the variable "Cutting-in-front" decreases the mean of "Helmet", implying that even if the collision is caused by another vehicle cutting in front, wearing a helmet

can decrease the risk of death for older riders. The variable "Alignment" increases the mean of "Helmet", indicating that, although wearing a helmet can reduce the risk of death for elderly riders, crashes on curved roads may increase the probability of an old rider being fatally injured. Lastly, the variable "Alignment" decreases the mean of "Violation". Since the mean of the random parameter "Violation" was not significant in the model, this interaction has no certain influence on the outcome severities besides the degree of randomness from the significant standard deviation. In terms of unobserved heterogeneity in variances, the variable "Work Zone" increases the variance of the random parameter "Helmet", thus making the distribution of the random parameter wider and increasing the variability and randomness of the random parameter's effect across the crash population.

### 5.3. Comparing Young and Old Rider Models

The marginal effect (ME) was used to compare the contributing factors between the young and old rider models and provide an overview of the severity changes for all observations [54] (Table 5 and Figure 4). The ME values for the models showed that older drivers were more likely to die (with fewer negative values) due to their weaker bodies and to potentially experience more difficult recovery compared to young drivers. This finding is consistent with the expectation that older individuals may be more vulnerable in the event of a crash.

**Table 5.** Marginal Effect.

| Contributing Factor | Marginal Effect | |
|---|---|---|
| | Young Rider | Old Rider |
| Rider characteristics | | |
| Gender | 0.1055 ** | 0.0676 ** |
| Foreigner | 0.0225 * | 0.0204 |
| Local address | −0.0628 ** | −0.0199 ** |
| Helmet | −0.0183 ** | −0.0779 ** |
| DUI | −0.0760 ** | −0.0797 ** |
| License status | −0.1243 ** | −0.0838 ** |
| Crash characteristics | | |
| Multi-Vehicle Crash | −0.0372 ** | −0.0484 ** |
| Rider at-fault | −0.0326 ** | −0.0146 ** |
| Exceeding PSL | 0.1354 ** | 0.0994 ** |
| Violation | | 0.0101 |
| Overtaking | | 0.0341 ** |
| Mobile use | −0.0963 * | −0.1251 ** |
| Asleep | 0.0617 ** | 0.0751 ** |
| Cutting-in-front | −0.0352 ** | |
| Environmental characteristics | | |
| Rain | −0.1076 ** | 0.0301 |
| Dust/foggy | −0.0707 ** | −0.0334 ** |
| Lit road | 0.0576 ** | 0.0287 ** |
| Unlit road | 0.0764 ** | 0.0423 ** |
| Road characteristics | | |
| Wet road | −0.0215 | −0.0444 ** |
| Work zone | 0.1108 ** | |
| Road surface | −0.1360 ** | −0.0156 ** |
| Alignment | 0.0170 ** | |

Note: * *p*-value < 0.1, ** *p*-value < 0.05.

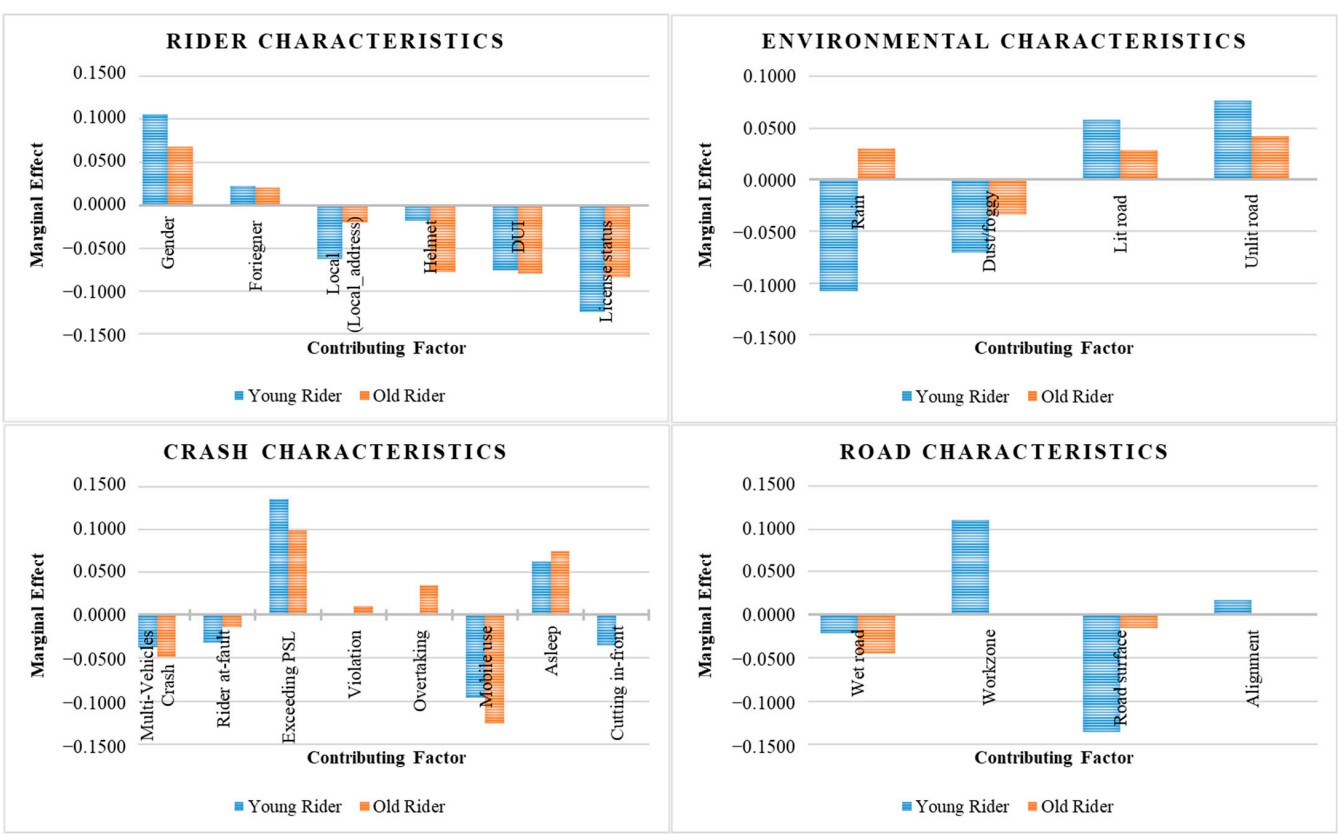

**Figure 4.** Marginal Effect.

The gender of the driver was found to have a similar impact on crash severity for both young and old rider models, with male drivers having more severe crashes than female drivers. This may be due to the fact that young male drivers, in particular, engage in risky behaviors, such as racing at night [70]. The foreign driver factor was insignificant in the old rider model, suggesting that foreign drivers in Thailand may have a higher risk of death, particularly among young drivers. Collisions involving local riders were found to have a low chance of death for both young and old riders, but the chance of death was lower for young riders. This may be due to their faster decision-making speed [49]. However, this variable was found to be a random parameter for both models, indicating that some groups of riders, even local residents, had a high risk of death.

The use of a helmet was found to reduce the chance of death for both young and old riders, though the old rider model showed this factor as a random parameter, possibly due to the use of non-standard helmets [71]. Crashes involving drunk drivers tended to have a lower severity for both young and old riders, possibly because these crashes typically involve short distances and few other vehicles, resulting in less severe outcomes. The license status was found to reduce the likelihood of death for both models, with young riders having a greater reduction in severity. This may be due to their faster decision-making speed compared to older riders [49].

Multi-vehicle crashes were found to have a lower chance of causing fatalities, possibly due to the slow speed of local road traffic. Crashes caused by motorcyclists were found to have a lower chance of causing death, with young riders having a lower chance of death than older riders. Exceeding the speed limit was found to have a high chance of causing death in both models, but young drivers had a much higher chance of death, according to the ME value. This may be due to the fact that young drivers commit more speeding violations and have more serious crashes [72]. The violation of traffic signals or other regulations was found to be insignificant in both models, but the old rider model was found to be a random parameter. Illegal overtaking was found to be significant only for

old riders, possibly because it often leads to head-on collisions that have a high chance of causing fatalities for elderly riders [40]. Crashes caused by the use of mobile phones while driving were found to have a lower chance of causing fatalities in both models. Crashes caused by falling asleep were found to increase the likelihood of death in both models, with falling asleep being particularly dangerous for motorcyclists due to the lack of safety equipment and inadequate braking capabilities [73]. Cutting in front was found to have fewer deaths for young riders only.

The environmental factor of rainy weather was found to have different impacts on crash fatalities for young and old riders. Young riders were found to have a negative significance (decreased crash fatalities), while old riders were found to have a positive significance (increased crash fatalities). This may be due to the fact that both young and old drivers will drive slowly in the rain, but older riders are more likely to die in a crash due to their weaker bodies and slower recovery time, as well as possibly slower rescue efforts. However, this factor was found to be a random parameter for young riders, which may indicate that some young riders have a higher chance of fatality in certain cases, such as when they are driving at high speeds and lose control. The lighting factor was found to increase the likelihood of mortality for both young and old riders, with young riders having a higher chance of death, according to the ME value. This may be due to the fact that young drivers tend to drive faster at night on open local roads and may not be able to brake in time if an animal suddenly runs in front of them in low-light conditions.

Crashes occurring on wet roads were found to have a lower chance of causing death for older riders, while this factor was found to be insignificant for young riders. This may be due to the fact that more experienced riders are more aware of the dangers of wet roads and take more caution while driving. Crashes occurring in areas under construction or maintenance were found to have a high chance of causing death, but only for young drivers. This may be due to the reckless driving behavior and inexperience of young drivers in recognizing warning signs, leading to higher chances of death in the event of a collision with an unexpected obstacle in a work zone [74]. The roughness of the road surface was found to be significantly negative for both models, indicating a lower chance of fatality when a crash occurs. However, this variable was a random parameter for both models, indicating that there were some crashes that caused a higher chance of death. The alignment variable revealed that if a young rider collided on a curve, it would result in a high chance of death. This may be due to the fast driving behavior of young riders, which may cause them to lose control while cornering at high speeds, while older riders tend to drive more slowly [75].

## 6. Conclusions and Implementations

This study aimed to identify the factors that affect the severity of motorcycle crashes on local roads by comparing the ages of the victims, including young riders under 30 years old and old riders over 50 years old. The focus on the severity of motorcycle crashes on local roads has not been previously explored, making this study a valuable contribution to the knowledge gap in this area. The findings of this study could potentially be used to make policy recommendations for reducing the severity of local road crashes, particularly in developing countries or low- to middle-income countries where motorcycles are a common mode of transportation. These recommendations could be implemented by various agencies, including the Department of Transport, which is responsible for training and issuing driver's licenses; public health agencies, such as the Bureau of Health Promotion; agencies that maintain local roads; and local government organizations.

This study analyzed crash data from 2015 to 2020 to identify factors that contribute to the severity of motorcycle crashes on local roads. These contributing factors were divided into four categories: driver, crash, environmental, and road factors. The analysis was conducted using a random parameter logit model with unobserved heterogeneity in means and variances. The transferability test revealed that the models for young riders and old riders were different, and it was appropriate to analyze them separately. The constant

values of the models indicated that old riders were more likely to die in a crash than young riders. In terms of the random parameter, the local address variable was found to be significant in both models, but the young rider model also included the rain variable, while the old rider model included the helmet and violation variables. Most of the random parameter values had the same direction, with the exception of rain, which had a different direction. The predominant marginal effect values were gender, exceeding the speed limit, lit road, unlit road, mobile use, and road surface.

In making policy recommendations, we considered factors such as the magnitude of the marginal effect on the probability of death and the feasibility of implementing various countermeasures. Based on the results of our model analysis, the following recommendations can be made for both models.

According to the findings of this study, male riders have a particularly high mortality rate in motorcycle crashes on local roads. To address this issue, it may be helpful to present statistical information on the risks of motorcycle accidents to raise awareness among riders of all genders and ages. Agencies such as the Department of Transport, which is responsible for issuing driver's licenses and providing training, could incorporate this information into their curricula. The results of the study also showed that having a driver's license can reduce the severity of crashes, indicating that driver training can be an effective means of improving road safety. However, it is important to ensure that training materials are up-to-date and include information about the latest best practices for safe driving [76]. To improve safety for foreign drivers in Thailand, it may be helpful to develop a platform that allows these individuals to obtain training and take a driving test before renting a vehicle. This could be implemented by agencies, such as the Ministry of Tourism and the Department of Land Transport. The study also found that wearing a helmet can significantly reduce the severity of motorcycle crashes on local roads, especially for older riders. To encourage more people to wear helmets, it may be helpful to conduct campaigns to raise awareness of the benefits of helmet use and provide information about helmet standards [77]. In urban areas, law enforcement agencies could also use tools, such as CCTV cameras, to enforce helmet laws and ensure that riders are protected [78].

Among the environmental factors considered in this study, crashes during rainy weather should be a particular concern for older drivers. To mitigate this risk, it is recommended to raise awareness of the hazards presented by rain, such as reduced visibility of road shoulders, through driver's license training and publicizing this information in places like village bulletin boards. To further reduce the risk of collisions, it may be advisable to minimize travel at night, as the presence of bright lights was found to have only a slight protective effect against fatal crashes. Since many nonstandard intersections and traffic conditions could be expected on a local road, a standard lighting provision could significantly improve the safety benefit [79], especially for vulnerable road users.

Single crashes, which were found to have a high probability of resulting in death in this study, may often be caused by drivers falling asleep at the wheel. To address this issue, safety engineers should identify locations where such crashes are likely to occur and implement countermeasures, such as the installation of flashing lights. However, it is important to note that the installation of pavement markers may not be effective in preventing these types of crashes, as they may cause riders to lose control of their vehicles and increase the risk of accidents [80]. Another significant factor identified in this study is crashes caused by exceeding the posted speed limit (PSL). To address this issue, it may be necessary to emphasize the dangers of fast driving through public health campaigns, public relations efforts, and driver education initiatives. This could include the use of public relations posters, media outreach, and incorporating information about the risks of speeding into driver training curricula. These efforts may be particularly important for young riders, who were found to have a higher mortality rate than older riders [81]. Since a previous study [82] found that converting minor-approach-only stop (MAS) intersections to all-way stop (AWS) intersections could potentially reduce the minimum speed in major

approaches up to 60%, this should also be given consideration for implementation to reduce operating speeds on local roads.

The road characteristics within a work zone have a significant impact on the behavior and safety of young riders, who may possess less experience and a tendency towards reckless driving. When encountering construction areas, these individuals are at an increased risk of fatal accidents. To mitigate this risk, it is recommended to implement measures such as raising awareness through education and publicizing the dangers of the construction zone; implementing traffic control measures, such as lane closures and detours; and installing physical barriers and warning signs to clearly communicate the presence of the work zone to all road users. Additionally, construction contractors and those responsible for ensuring construction standardization should consider implementing measures to ensure that appropriate warning signs are placed at appropriate distances to alert road users of the construction zone.

## 7. Limitation and Further Research

This study represents one of the first comprehensive examinations of motorcycle crashes on local roads in a national context, and its findings may be applicable to other developing countries where motorcycles are also commonly used. However, due to the diversity of driving behaviors present in these contexts, further research is necessary to fully understand the factors contributing to motorcycle crashes [83]. This could include spatial analysis and studies that examine differences in behavior between genders or age groups using naturalistic riding principles [84]. To effectively address the issue of motorcycle crashes in these contexts, it may be necessary to conduct focus groups and in-depth interviews to better understand the unique challenges and opportunities present in these locations [4]. An important limitation is that the current study applied an advance heterogeneity model that takes into consideration the temporal stability investigation across time periods, which has been widely observed in numerous crash-related injury severity studies [85]. Since Thailand began the COVID-19 lockdown in early April 2020, instability during the last several months may be even more likely, as found in numerous crash severity studies during the COVID-19 period [86]. Therefore, this issue is recommended for the direction of future research. Lastly, some important factors related to land use (e.g., agricultural, residential, industrial, or recreational use, etc.) should also be considered in future studies, since local roads and land use may potentially have great influence on the outcome severity.

**Author Contributions:** Conceptualization, T.C.; methodology, T.C.; software, T.C. and V.R.; validation, C.S. and T.B.; formal analysis, T.C.; investigation, T.C. and C.S.; data curation, T.B.; writing—original draft preparation, T.C. and C.S.; writing—review and editing, T.C. and C.S.; visualization, S.J.; supervision, V.R.; project administration, S.J.; funding acquisition, T.C. All authors have read and agreed to the published version of the manuscript.

**Funding:** This research project is supported by the Science Research and Innovation Fund [Grant No. FF66-P1-015].

**Institutional Review Board Statement:** This research was approved by the Ethics Committee for Research Involving Human Subjects, Rajamangala University of Technology, Isan (HEC-01-65-078).

**Informed Consent Statement:** Not applicable.

**Data Availability Statement:** The data presented in this study are available on request from the corresponding author. The data are not publicly available due to privacy policy restrictions.

**Acknowledgments:** The authors would like to thank the Department of Disaster Prevention and Mitigation, Ministry of Interior, for supporting the road traffic crash data.

**Conflicts of Interest:** The authors declare no conflict of interest.

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
