# Peer review of "A Comparison of Contributing Factors between Young and Old Riders of Motorcycle Crash Severity on Local Roads"

_sustainability, doi:10.3390/su15032708_

Round 1
Reviewer 1 Report
The study used a random parameter logit model with unobserved heterogeneity in means and variances to compare the crash severity contributing factors between young and older motorcycle riders. This is an interesting and important topic that is worth considering. The modeling of the article is very good and accurate and well presented. However, I would like the authors to address the following minor issues before publication.
Developing countries are rightly mentioned in the keywords. I suggest you say in the abstract or title of the article which country the study is for.
In the abstract, the young riders are (30-year-old); might be <30-year-old or =<30-year-old.
Since the article is related to developing countries, I recommend to strengthen the literature review as well as the discussion section by considering the findings of the articles that have been recently published in the developing countries in other regions (e.g., Iran, India or Pakistan). The comparison between the findings in developing countries and then comparison with developed countries can enrich the discussion.
The time period was from 2015 18 to 2020. Many studies have shown that Covid-19 has affected the number and severity of crashes. How is this temporal instability included in your study?
More clarification is required about the findings related to the unobserved heterogeneity in variances. What is your logic in interpreting the effect of Violations on the variance of the Rain variable? Why does this effect increase the probability of a fatal accident for only 37.8% of people? Similar to this type of interpretation is also done in line 492, which leads to the same question. (line 445, the percentage might be 32.7 not 32.3)
Author Response
Respond to evaluation by Reviewer#1:
The study used a random parameter logit model with unobserved heterogeneity in means and variances to compare the crash severity contributing factors between young and older motorcycle riders. This is an interesting and important topic that is worth considering. The modeling of the article is very good and accurate and well presented. However, I would like the authors to address the following minor issues before publication.
Comment: Developing countries are rightly mentioned in the keywords. I suggest you say in the abstract or title of the article which country the study is for.
Authors response: we thank you for the suggestion. we have now added the phrase “in Thailand-a middle-income developing country” in the abstract part of our paper. (Please see Line 19)
Comment: In the abstract, the young riders are (30-year-old); might be <30-year-old or =<30-year-old.
Authors response: Thank you for pointing it out. We have now revised it to be “< 30-year-old”. (See Line: 20)
Comment: Since the article is related to developing countries, I recommend to strengthen the literature review as well as the discussion section by considering the findings of the articles that have been recently published in the developing countries in other regions (e.g., Iran, India or Pakistan). The comparison between the findings in developing countries and then comparison with developed countries can enrich the discussion.
Authors response: We thank you for the insightful suggestion. Agreeing with your suggestion, we have now added several research studies in the context of the developing countries (including Vietnam, Iran, India, Pakistan, and Ghana) into the Table 1 (Highlighted in yellow) and we have provided brief explanation for why research in the context of Thailand may potentially different from the previous studies due to spatial instability. (See Line: 75-82).
We also provided more comprehensive discussion of our research result, compared with the finding of the research from other developing countries. Please see the yellow highlighted part in Sub-section 5.1 Young rider model and 5.2 Old rider model.
Comment: The time period was from 2015 to 2020. Many studies have shown that Covid-19 has affected the number and severity of crashes. How is this temporal instability included in your study?
Authors response: We thank you for your insightful question. We are aware that numerous studies have found that the Covid-19 have cause some instability in both factors influencing the severity and crash rate. In the case of Thailand, the Covid-19 lockdown began in April of 2020 (only between 10 PM to 5 AM). Therefore, only 9 months of the current study’s data were in the period of lockdown and only 7 hours daily during the nighttime. However, it is not deniable that the effect of the explanatory variable could change overtime, not just because of the Covid-19 but also other reasons as review in by Mannering (2018). This is one of the limitation of the current paper that we would like to recommend for further studies. Please refer to the limitation discuss part of the paper (Line 730-736).
Comment: More clarification is required about the findings related to the unobserved heterogeneity in variances. What is your logic in interpreting the effect of Violations on the variance of the Rain variable? Why does this effect increase the probability of a fatal accident for only 37.8% of people? Similar to this type of interpretation is also done in line 492, which leads to the same question. (line, 445, the percentage might be 32.7 not 32.3)
Authors response: We thank you for your question for clarification. First, we would like to respond to this question “What is your logic in interpreting the effect of Violations on the variance of the Rain variable? Why does this effect increase the probability of a fatal accident for only 37.8% of people? Similar to this type of interpretation is also done in line 492, which leads to the same question.”. The interpretation in the previous version of is not accurate and correct. The influence of the exogenous variable on the variances of the random parameters can only decrease or increase the variability or randomness of the effect (coefficient for each crash population) of the random parameters (in our study, normal distribution was applied), thereby making the distribution of the random parameters smaller or wider, respectively. Therefore, we would like to revise our interpretation as follows:
Line: 472-482:
“Unobserved heterogeneity in variances was also found to be a significant predictor. That is, the crashes involving traffic sign/signal/wrong-direction violation increase the variance of crashes occurring during raining. Such increase in variances of the random parameter make its’ distribution wider and increases their randomness. In simple term, the “Violation” makes the parameter estimate of each crash observation of the random parameter “Rain” even more varied or random across crash populations, compared to the degree of randomness generated by the random parameters without considering the variances heterogeneity) [66]. This, to some degree, captured the effect of unmeasured factors or characteristics related to the detailed information on variable reflecting riders’ violation which may go beyond the police report capabilities, thus making the accuracy of the crash modelling more reliable [67-69]”
Line: 544-577
“In terms of unobserved heterogeneity in means, the variable “Cutting-in-front” increases the mean of “Local address”, thus making the risk of death more likely for old riders. In contrast, the variable "Cutting in-front" decreases the mean of “Helmet”, implying that even if the collision is caused by another vehicle cutting in front, wearing a helmet can de-crease the risk of death for older riders. The variable "Alignment" increases the mean of “Helmet”, indicating that, although wearing a helmet can reduce the risk of death for elderly riders, crashes on curves road may increase the probability of old rider being fatally injured. Lastly, the variable "Alignment" decreases the mean of “Violation”. Since, the mean of random parameter “Violation” was not significant in the model, this interaction has no certain influence on the outcome severities beside the degree of randomness from the significant standard deviation. In terms of unobserved heterogeneity in variances, the variable "Work Zone" increases the variance of the random parameter “Helmet”, thus making the distribution of the random parameter wider and increasing the variability and randomness of the random parameter’s effect across crash population.”
Secondly, we have now corrected the distributional percentage of the helmet random parameter from 32.3% to 32.7% as suggested. (see Line 500)
Reviewer 2 Report
Please revise the literature review for instance "This has led to efforts to halve 35 road deaths as part of the Sustainable Development Goals (SDG) Target 3.6" 3.6 is not clear.
As the title mentioned local roads, the lack of intersections and traffic situations especially during day and night are felt. following papers discussed this issue.
Standard against nonstandard urban intersection nighttime traffic safety evaluation using cross-sectional method
-
https://doi.org/10.1007/s41062-021-00569-y
- or
- Do stop-signs improve the safety for all road users? A before-after study of stop-controlled intersections using video-based trajectories and surrogate measures of safety
- https://doi.org/10.1016/j.aap.2021.106563
Please add more description regard of land use since the subject focused on local roads and land use has a great impact on these categories.
Author Response
Respond to evaluation by Reviewer#2:
Comment: Please revise the literature review for instance "This has led to efforts to halve road deaths as part of the Sustainable Development Goals (SDG) Target 3.6" 3.6 is not clear.
Authors response: We thank you for your comment. We have clarified this part as follows (Line 35-38):
“Due to detrimental effects on global health systems and economy, United Nation has developed Targe 3.6 aiming to reduce half of road traffic deaths by 2030 as part of the Sustainable Development Goals (SDG) [2]”
Comment: As the title mentioned local roads, the lack of intersections and traffic situations especially during day and night are felt. Following papers discussed this issue:
Standard against nonstandard urban intersection nighttime traffic safety evaluation using cross-sectional method: https://doi.org/10.1007/s41062-021-00569-y
Do stop-signs improve the safety for all road users? A before-after study of stop-controlled intersections using video-based trajectories and surrogate measures of safety: https://doi.org/10.1016/j.aap.2021.106563
Authors response: We thank you for your suggested paper. After carefully review them, we are able to cite them in our policy and recommendation to improve safety for motorist on local road as follows:
Line 687-689 (recommendation regarding lighting variable):
“Since many nonstandard intersections and traffic condition could be expected on the local road, standard lighting provision could significantly improve the safety benefit [82], especially for vulnerable road users.”
Line 702-706 (recommendation regarding exceeding speed variable):
“Since previous study [85] found that converting minor-approach-only stop (MAS) inter-sections to all-way-stop (AWS) intersection could potentially reduce the minimum speed in the major approaches up to 60%, this should also be given a consideration for implication to reduce operating speed on the local road.”
Comment: Please add more description regard of land use since the subject focused on local roads and land use has a great impact on these categories.
Authors response: We thank you for your recommendation. We agree with you that lane-use related variables may potentially be statistically significant in influencing the outcome severity. However, those related-variables details are not available in our current crash data set in Thailand. Therefore, we would like to address this issue as one of the limitation of our study. (See Line 736-739)
Reviewer 3 Report
I am happy with the current version of the manuscript.
Only minor spell check is needed.
Author Response
Respond to evaluation by Reviewer#3:
Comment: I am happy with the current version of the manuscript. Only minor spell check is needed.
Authors response: We thank you for your time and consideration and positive feedback on our manuscript. We have now double-checked for minor typo, error, grammar mistake etc., through out the manuscript.